# Emerging Role of Natriuretic Peptides in Diabetes Care: A Brief Review of Pertinent Recent Literature

**DOI:** 10.3390/diagnostics14192251

**Published:** 2024-10-09

**Authors:** Dipti Tiwari, Tar Choon Aw

**Affiliations:** 1ResteLab, Singapore 318877, Singapore; tiwari.dipti02@gmail.com; 2Department of Laboratory Medicine, Changi General Hospital, Singapore 529889, Singapore; 3Yong Loo Lin School of Medicine, National University of Singapore (NUS), Singapore 119228, Singapore; 4Pathology Academic Clinical Program, Duke-NUS Graduate School of Medicine, Singapore 169857, Singapore

**Keywords:** BNP, diabetes, heart failure, natriuretic peptides, NT-proBNP

## Abstract

Diabetes markedly increases susceptibility to adverse cardiovascular events, including heart failure (HF), leading to heightened morbidity and mortality rates. Elevated levels of natriuretic peptides (NPs), notably B-type natriuretic peptide (BNP) and N-terminal-proBNP (NT-proBNP), correlate with cardiac structural and functional abnormalities, aiding in risk stratification and treatment strategies in individuals with diabetes. This article reviews the intricate relationship between diabetes and HF, emphasizing the role of NPs in risk assessment and guiding therapeutic strategies, particularly in individuals with type 2 diabetes mellitus (T2DM). We also explore the analytical and clinical considerations in the use of natriuretic peptide testing and the challenges and prospects of natriuretic-peptide-guided therapy in managing cardiovascular risk in patients with diabetes. We conclude with some reflections on future prospects for NPs.

## 1. Introduction

Diabetes mellitus is a global health challenge associated with alarming rates of morbidity and mortality. As of 2021, the worldwide count of diabetes patients reached a staggering 529 million; this figure is projected to escalate exponentially to over 1.3 billion by 2050 [1]. In addition to its pervasive prevalence, diabetes poses a significant threat to affected individuals with respect to adverse cardiovascular events and heightened hospitalization rates when compared to their non-diabetic counterparts. There is now growing recognition of the importance of the macrovascular complications of diabetes in addition to the traditional focus on microvascular complications (retinopathy, neuropathy, and nephropathy) [2]. Type 2 diabetes mellitus (T2DM) increases the susceptibility to heart failure (HF) by directly impairing cardiac function as well as through associated comorbidities like hypertension, obesity, metabolic disorder, and renal dysfunction [3]. The 1974 Framingham Heart Study (*n* = 5209), which had an 18-year follow-up, underscores the fact that diabetes independently amplifies the risk of HF by 2-fold in men and an astonishing 4-fold in women compared to age-matched controls [4]. A more recent report highlighted that, amongst 3.25 million people in Scotland (>30 years old), the incidence of heart failure hospitalization in those with T2DM was 5.6/1000 person years versus 2.4 in those without [5]. In patients with T2DM, the prevalence of HF ranges from 9% to 22% and even higher in those aged >60 years [6]. In addition, several studies have demonstrated worse outcomes of HF in patients with diabetes. Diabetes accelerates the progression of HF from preclinical stages to more advanced stages [7]. Given the significant increase in the global prevalence of diabetes over the past decade, and with further increases anticipated in the years ahead, the burden of HF on healthcare systems can only grow [7]. Accordingly, over the recent years, risk stratification tools have been proposed to estimate HF incidence and to guide treatment in those with diabetes [8]. In fact, the American Diabetes Association 2022 guidelines highlight the advantages of incorporating HF biomarkers in the risk stratification of patients with diabetes [9]. In this regard, there is a growing body of evidence for using natriuretic peptides (NP) either as a single variable or as a part of multivariate models for predicting HF in patients with T2DM.

In this review, we aim to summarize the pertinent knowledge about the current use of NPs in the risk stratification and management of patients with T2DM. In addition, we explore the analytical and clinical considerations in the use of natriuretic peptide testing and the challenges and prospects of natriuretic-peptide-guided therapy in managing cardiovascular risk in patients with diabetes.

## 2. Pathophysiology of Heart Failure in Diabetes

There has been a recognized universal definition of HF since 2021 [10]. This states that HF is a clinical syndrome with symptoms and/or signs caused by a structural and/or functional cardiac abnormality and corroborated by elevated NP levels and/or objective evidence of pulmonary or systemic congestion. It is characterized by the inability of the heart to pump enough blood and oxygen to support the metabolic needs of the body [11]. Thus, based on the left ventricular ejection fraction (LVEF), the latest guidelines have also classified HF accordingly: HF with reduced ejection fraction (HFrEF; LVEF ≤ 40%), HF with mildly reduced ejection fraction (HFmrEF; LVEF 41–49%), and HF with preserved ejection fraction (HFpEF; LVEF ≥ 50%). A subcategory of HF with improved ejection fraction (HFimpEF) is added if a baseline LVEF ≤ 40% shows a ≥10-point increase on a second measurement, i.e., LVEF > 40% [10]. Though the etiopathology of HF is multifactorial, it is largely influenced by structural and cellular alterations in the presence of chronic comorbid conditions, such as hypertension, diabetes, pulmonary or renal insufficiency, and cancer [12].

Diabetes remains a major risk factor for the development of HF. Similarly, HF significantly contributes to the progression and worsening of diabetes. The complex interplay between HF and T2DM encompasses a multitude of molecular, cellular, and systemic mechanisms. At the core of their bidirectional relationship lie hyperglycemia and insulin resistance, influenced by common risk factors such as obesity and inflammation [13]. Diabetes is marked by neurohormonal dysregulation and the activation of the renin–angiotensin–aldosterone system (RAAS), culminating in heightened oxidative stress and inflammation at the endothelial level. This cascade of events results in endothelial dysfunction, a pivotal factor contributing not only to impaired cardiac contractility but also to the initiation and progression of atherosclerosis, thereby exacerbating the development of HF [14]. In addition, the presence of myocardial lipotoxicity and diabetic cardiomyopathy, characterized by fibrosis and altered energetics, creates a milieu that predisposes individuals to HF [13]. Furthermore, in the context of diabetes, there is evidence suggesting that hyperactivation of platelets and coagulation dysfunction may contribute to the worsening of HF, adding another layer of complexity to the pathophysiological processes linking these two conditions [15]. This intricate web of interconnected mechanisms underscores the importance of a comprehensive understanding for the development of targeted therapeutic strategies to mitigate the risk and progression of HF in individuals with diabetes.

## 3. Natriuretic Peptides and Their Clinical Applications in Heart Failure

Natriuretic peptides are a family of hormones that play a crucial role in regulating blood pressure, fluid balance, and cardiovascular health. There are three biologically active peptides in the NP family: atrial natriuretic peptide (ANP), BNP, and C-type natriuretic peptide (CNP) [16]. First discovered in 1980 by De Bold, initially from the atria of the heart, ANP was found to have potent diuretic and natriuretic effects, leading to increased urinary output and sodium excretion [17]. Shortly thereafter, BNP and CNP were isolated from the brain, although their major production sites were later found to be the ventricles of the heart and endothelial cells, respectively [18]. Among these three NPs, BNP stands out for its pivotal role in maintaining cardiovascular homeostasis. It is primarily synthesized and secreted by the ventricles of the heart in response to increased pressure or volume overload [19]. Once released into the bloodstream, BNP exerts its diverse effects by binding to specific receptors on target cells. These effects include natriuresis, diuresis, vasodilation, anti-inflammatory and anti-fibrotic effects, and antihypertrophic effects [20]. N-terminal pro-B-type natriuretic peptide (NT-pro BNP) is an inactive precursor molecule of BNP; it is cleaved from pro-BNP to produce the active BNP molecule. Like BNP, NT-pro BNP levels are elevated in response to increased myocardial stress [21].

The American Heart Association (AHA) recognizes the crucial role of BNP and its precursor NT-pro BNP in both diagnosing and risk stratifying HF [22]. The current guidelines emphasize that elevated NT-pro BNP (>125 pg/mL) and BNP (>35 pg/mL) levels provide valuable evidence of the structural and functional alterations of the heart, thus aiding in the diagnosis of HF in both acute and chronic settings. In patients presenting with dyspnea, elevated BNP or NT-pro BNP levels assist in the determination of the need for further evaluation using transthoracic echocardiography. Furthermore, proactive screening based on BNP or NT-pro BNP levels, along with collaborative care involving cardiovascular specialists, proves beneficial for those at risk of developing HF, aiding in the prevention of left ventricular dysfunction or the onset of new HF.

In the context of chronic HF, it is recommended to regularly measure BNP or NT-pro BNP levels for effective risk stratification. The Asia Pacific Society of Cardiology (APSC) developed consensus recommendations for managing chronic HF in the Asia–Pacific region in 2023, acknowledging regional disparities in HF epidemiology and healthcare resources [23]. Using the Grading of Recommendations Assessment, Development and Evaluation (GRADE) system, experts reached a consensus on diagnosis and treatment strategies, emphasizing the significance of NP levels in HF diagnosis. The 19-member expert panel agreed that a plasma concentration of NT-pro BNP < 125 pg/mL or BNP < 35 pg/mL renders a diagnosis of HF unlikely.

The strong correlation between elevated NP levels and the severity of HF underscores the potential utility of NP-guided therapy. In the EMPEROR-Reduced trial, associations between NT-pro BNP concentrations and the effects of treatment with empagliflozin, a sodium–glucose cotransporter 2 inhibitor (SGLT2i), were evaluated in patients with chronic HFrEF [24]. Empagliflozin demonstrated significant reductions in NT-pro BNP concentrations and HF events. Moreover, post-treatment NT-proBNP concentrations following empagliflozin therapy better informed subsequent prognosis compared to pretreatment concentrations [24].

Recently, the STRONG-HF trial examined the efficacy of intensive treatment involving rapid up-titration of guideline-directed medication and close follow-up post acute HF admission. Involving 1078 patients across 87 hospitals in 14 countries, the study found that high-intensity care significantly increased the proportion of patients up-titrated to full drug doses by day 90 and led to greater reductions in key HF indicators like NT-proBNP levels. Notably, high-intensity care substantially reduced the risk of 180-day all-cause death or HF readmission compared to usual care. These findings underscore the importance of close NT-proBNP monitoring in optimizing HF management strategies [25].

The 2024 REVOLUTION-HF (Real World Education in HF) study from Sweden reported on patients with suspected de novo heart failure presenting in outpatient settings [26]. One of its key findings was that, while NT-proBNP levels were quickly tested, there was a significant delay in confirming a HF diagnosis, largely due to long waits for echocardiography. After one year, only 29% of the patients received an HF diagnosis, which contributed to high rates of HF hospitalization and all-cause mortality, particularly in those with elevated NT-proBNP levels. In addition, the uptake of guideline-directed medical therapies (GDMT), such as beta-blockers, renin–angiotensin system inhibitors, and mineralocorticoid receptor antagonists, was limited in the first year of diagnosis. The study highlighted the need for faster initiation of life-saving therapies and a more pragmatic approach, advocating for an NT-proBNP-based rule-in strategy to diagnose HF earlier, without waiting for echocardiography [26].

## 4. Natriuretic Peptides for HF Risk Stratification and Management in Diabetes

People with diabetes face an increased risk of developing HF, a trend evident in numerous longitudinal observational studies. The heightened risk for HF in diabetes is categorized as stage A, denoting an increased risk without symptoms, structural heart disease, or biomarker evidence of myocardial strain. Those in stage B are asymptomatic but exhibit structural heart disease or functional cardiac abnormalities, making them particularly susceptible to progressing to symptomatic stages C and D [27]. To mitigate the progression to symptomatic HF, early identification, risk stratification, and treatment of risk factors are paramount. Notably, in T2DM, measuring BNP or NT-pro BNP proves valuable for identifying individuals at risk and predicting HF development, symptom progression, and HF-related mortality. Several randomized controlled trials have underscored the benefits of more intensive risk factor treatment guided by NP levels. Major findings from some of these studies are summarized below in Table 1.

The St Vincent’s Screening to Prevent Heart Failure (STOP-HF) study involved 1374 participants at HF risk (including those with diabetes), randomized into usual care or BNP screening groups. Those with BNP ≥ 50 pg/mL received echocardiography and collaborative care. The primary outcome showed a lower prevalence of LV dysfunction in the intervention group (5.3% vs. 8.7%) after a mean follow-up period of 4.2 years. Moreover, the incidence of cardiovascular hospitalizations was lower in the intervention group. BNP-based screening reduced the composite endpoint of incident asymptomatic LV dysfunction with or without newly diagnosed HF [28].The NT-proBNP Selected Prevention of Cardiac Events in a Population of Diabetic Patients without a History of Cardiac Disease (PONTIAC) study included 300 patients with T2DM and NT-proBNP > 125 pg/mL but without any history of cardiac disease. Participants were randomized into a control group (receiving usual care) and an intensified group, which received additional cardiac outpatient care for the up-titration of RAAS antagonists and beta-blockers and was followed up for 2 years. Notably, intensified therapy led to a significant reduction in cardiac hospitalizations/deaths compared to the control group (hazard ratio: 0.351; *p* = 0.044) [29].In the Examination of Cardiovascular Outcomes with Alogliptin versus Standard of Care (EXAMINE) trial, two NT-pro BNP measurements (6 months apart) in patients (*n* = 5380) with T2DM effectively identified those at the highest risk of developing symptomatic HF. Patients with persistently high or increasing NT-pro BNP levels at 6 months had a significantly higher risk of CV death or HF compared to those with consistently low levels or initial high levels that declined [30].The Canagliflozin Cardiovascular Assessment Study (CANVAS) involved 4330 participants with T2DM and either CVD or other risk factors for cardiac events. Plasma NT-proBNP concentrations were measured at baseline, year 1, and year 6, and associations between NT-proBNP levels and various cardiovascular, renal, and mortality outcomes were investigated. NT-pro BNP levels ≥ 125 pg/mL predicted incident hospitalization for HF (hazard ratio 5.40; *p* < 0.001). Furthermore, it was suggested that elevated NT-pro BNP levels could predict a wide range of deleterious cardiovascular and renal outcomes in T2DM, HF death, and all-cause mortality [31].The Thousand and I study assessed the prognostic significance of elevated NT-pro BNP levels in patients (*n* = 960) with type 1 DM with preserved LVEF and without known heart disease. During a median follow-up of 6.3 years, 121 participants experienced major cardiovascular events (MACE) and 51 died. Higher NT-pro BNP levels were linked to poorer outcomes, with adjusted hazard ratios for MACE at 1.56 and 4.29 per Loge increase for NT-proBNP [32].

Considering all this compelling evidence, the 2024 American Diabetes Association (ADA) practice guidelines recommend screening asymptomatic adults with diabetes using NPs, with a threshold for abnormal values set at BNP ≥ 50 pg/mL and NT-pro BNP ≥ 125 pg/mL [33]. Elevated natriuretic peptide levels, indicative of stage B HF, warrant further evaluation through echocardiography to screen for structural heart disease and diastolic dysfunction. A collaborative, interprofessional approach involving cardiovascular specialists is recommended at this stage to implement guideline-directed medical treatment strategies, potentially averting progression to symptomatic stages of HF. The 2023 European Society of Cardiology (ESC) guidelines recommend that patients with diabetes should be regularly monitored for symptoms and signs of HF (such as dyspnea, fatigue, and peripheral edema), and, in the presence of one or more symptoms, NPs should be measured to identify those with suspected HF [34]. The thresholds for abnormal values of BNP and NT-pro BNP are ≥ 35 pg/mL and ≥ 125 pg/mL, respectively. Accordingly, the diagnostic algorithm for HF in patients with diabetes can be summarized as shown in Figure 1.

With advances in pharmacotherapy over the past decades, the importance of early identification of HF has become even more critical. Early diagnosis allows for the prompt initiation of effective therapies that can slow disease progression, reduce hospitalizations, and improve overall patient outcomes. As treatments evolve, identifying HF in its early stages ensures that patients can benefit fully from these therapeutic innovations [35]. Numerous studies are currently underway to determine the most effective approach for identifying individuals at the highest risk of developing HF. A recent meta-analysis comprising 30 studies from clinical trials and cohorts (*n* = 44,889) aimed to assess the predictive utility of serum NT-proBNP levels in T2DM complications and reported a moderate level of evidence for predicting cardiovascular and all-cause mortalities at NT-proBNP levels of >100 and >225 pg/mL, respectively [36]. Two risk scores, namely the WATCH-DM (Weight, Age, Hypertension, Creatinine, High-Density Lipoprotein Cholesterol, Diabetes Control, ECG QRS Duration, Myocardial Infarction, and Coronary Artery Bypass Grafting) and TRS-HF_DM_ (Thrombolysis in Myocardial Infarction Risk Score for Heart Failure in Diabetes Mellitus), have been developed specifically to predict the risk of HF in individuals with type 2 diabetes [37]. In a recent study by Patel, selective NT-proBNP testing guided by the WATCH-DM score demonstrated efficiency in identifying a primary prevention population with diabetes (*n* = 6293) at high risk of developing HF [38]. A two-step screening approach, integrating NT-pro BNP testing, was suggested as an effective method to identify high-risk patients who stand to benefit most from treatment with SGLT2i. The optimal NP cut-off levels for ruling in and ruling out HF are summarized in Table 2.

## 5. Analytical and Clinical Considerations in Natriuretic Peptide Testing

Although the measurement of NP levels is a crucial aspect of HF diagnosis, there are inherent challenges in their laboratory analysis and clinical interpretation that necessitate careful consideration. First, adherence to specific sample requirements and meticulous laboratory analysis procedures is paramount to ensure accurate results. In particular, BNP is prone to proteolytic degradation and is only stable in whole blood for approximately 4 h at room temperature [39]. Moreover, for BNP analysis, only ethylenediaminetetraacetic acid (EDTA) plasma or EDTA-containing whole blood samples can be used. In this regard, NT-pro BNP offers an advantage as it is less vulnerable to degradation, can be analyzed in serum or heparin plasma, and is not significantly affected by moderate hemolysis or freeze–thaw cycles [40].

While NPs are regarded as the “gold standard” biomarkers for estimating HF risk, it is essential to recognize that elevated levels of these peptides can occur in conditions other than HF. Conditions such as acute coronary syndrome, chronic kidney disease, pulmonary embolism, sepsis, and atrial fibrillation can lead to elevated levels of NPs. Furthermore, factors such as age, gender, and body mass index can also influence NP levels, complicating their interpretation [10]. In addition, the utility of these peptides in HFpEF is subject to debate as their levels tend to be lower in HFpEF compared to HFrEF [41]. However, current guidelines do not recommend separate NP cut-offs for HFpEF, and the prognosis is similar in patients for a given NP concentration, regardless of LVEF [20].

The effects of medications on circulating levels of NPs are well recognized and users should be cognizant of them. For instance, most drugs used in treating HF, such as SGLT2is, diuretics, RAAS blockers, and dopamine agonists, may lead to lower NP levels. On the other hand, beta-blockers, digitalis, and aspirin may elevate the circulating levels of BNP and NT-pro BNP [41,42]. Neprilysin is an endopeptidase responsible for the lysis of NPs. Neprilysin inhibition (through the use of angiotensin receptor neprilysin inhibitors or ARNIs) may have variable effects on natriuretic peptide assays. Thus, caution is warranted when interpreting BNP results in patients on ARNIs [43].

The best use of NPs is to support clinical judgment in the diagnosis of acute HF where NPs are markedly elevated. They are also useful in the exclusion of HF when NPs are very low. It is important to recall that the pathophysiology of HF is complex and may involve many processes such as myocardial stretch/stress, inflammation, fibrosis, remodeling, oxidative stress, and neuroendocrine effects [44]. It is tempting to consider the use of other cardiac biomarkers, e.g., soluble source of tumorigenicity 2 (sST2), growth differentiation factor-15 (GDF-15), and mid-regional pro-atrial natriuretic peptide (MR-proANP), to improve personalized care [45]. While helpful in some instances, these biomarkers remain largely promissory [44]. Currently, NPs remain as the gold standard for HF management [45].

Race and ethnicity are known demographic parameters affecting NP levels in healthy individuals. Racial and ethnic minorities have the highest burden of HF [46]. Black/Sub-Saharan or Hispanic individuals have increased risk of hypertension or HF compared to Caucasians or Asians [47] despite NT-proBNP levels being lower among Black and Mexican American adults compared to White adults [48]. The interpretation of racial and ethnic differences in circulating NPs is unclear and merits more studies in different populations. It is noteworthy that a panel of experts concluded that there are no significant diagnostic or prognostic differences in HF between Asian and Western populations [49]. It is also noteworthy that, in the diagnosis of acute HF, NTproBNP levels in patients from Singapore were slightly more accurate than in those from New Zealand [50]. This was attributed to the younger age of the HF cohort from Singapore. In another study NT-proBNP was equally prognostic for HF patients in Singapore and Caucasian subjects [51]. The racial/ethnic HF disparities may also be due, in part, to social determinants of health and inequities in access, therapy utilization, and resource allocation [46].

## 6. Future Prospects and Conclusions

B-type natriuretic peptides (BNP and NT-proBNP) have emerged as potential biomarkers in cardiovascular risk assessment and stratification of patients with diabetes. Additionally, natriuretic-peptide-based therapeutics represent another compelling area of research interest. ARNIs, such as sacubitril/valsartan, have shown efficacy in reducing cardiovascular events and mortality in patients with HF, including those with T2DM [52]. In addition to their cardiovascular benefits, ARNIs may also impact glucose homeostasis in individuals with diabetes. While some studies have suggested potential improvements in glycemic control with ARNI therapy, others have reported conflicting findings or no significant effect on glucose metabolism [53]. The mechanisms underlying the effects of ARNIs on glucose homeostasis are not fully understood and may involve complex interactions within the RAAS, NP system, and other pathways influencing insulin sensitivity and glucose metabolism.

Natriuretic-peptide-guided treatment of HF in patients with diabetes has gained increasing attention due to its potential to improve outcomes by optimizing therapy. The findings from major clinical trials (summarized in Table 3) demonstrate the importance of incorporating NP levels in guiding therapy for HF; where this was not carried out, the results were underwhelming. Even in patients with HFpEF, a more challenging condition that is prevalent in diabetes, recent trials like EMPEROR-Preserved and DELIVER using NP-tailored treatments showed improved cardiovascular outcomes.

While concerns exist that the recognition of HF risk through NP testing might trigger further tests, potentially increasing costs and management complexity, the high negative predictive value of NPs mitigates this by effectively ruling out individuals at minimal risk, thus reducing unnecessary interventions [2,60]. Furthermore, early detection of HF risk prompts timely intervention, potentially improving outcomes. Walter demonstrated the cost effectiveness of NP-based screening in patients with diabetes, underscoring its importance in bridging the diagnostic and treatment gap in HF care, especially when intervention is most impactful [61]. In addition, a recent study reported the association between plasma microRNAs (miRNAs) and cardiac biomarkers (such as NPs), along with their correlation with cardiac structure, function, and clinical outcomes in patients with symptomatic HF. The study found significant associations between miRNAs and biomarkers with ventricular size and function [62]. These findings indicate that future research may focus on integrating information from miRNAs with NPs to enhance prognostic assessment in diabetic cardiomyopathy beyond conventional methods.

Overall, NP-guided therapy represents a promising approach for managing cardiovascular risk, especially heart failure, in individuals with diabetes. More widespread and early adoption of NP-guided protocols can only improve outcomes. In addition, more studies in different population groups, ethnicities, and geographic regions using intensive guideline-directed medical therapy incorporating all four drug pillars including SGLT2i are needed. Further research is also warranted to elucidate the precise mechanisms of action and the optimal use of therapies in diabetic populations. Additionally, close monitoring of glycemic control and collaboration between healthcare providers specializing in cardiology and endocrinology are essential to ensure comprehensive management of cardiovascular and metabolic health in patients with diabetes.

## Figures and Tables

**Figure 1 diagnostics-14-02251-f001:**
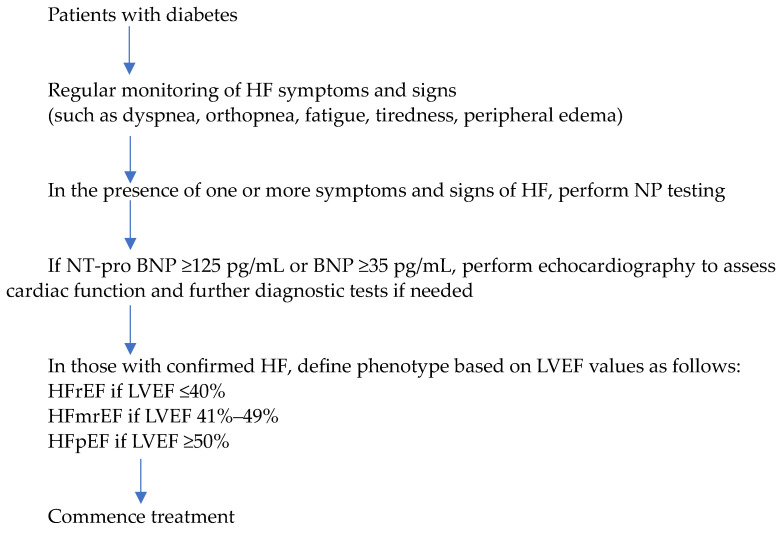
Diagnostic algorithm for heart failure in patients with diabetes. BNP, B-type natriuretic peptide; HF, heart failure; HFmrEF, heart failure with mildly reduced ejection fraction; HFpEF, heart failure with preserved ejection fraction; HFrEF, heart failure with reduced ejection fraction; LVEF, left ventricular ejection fraction.

**Table 1 diagnostics-14-02251-t001:** Major studies on NPs for HF risk stratification in diabetes.

Cohort Name	Population Studied	Follow-Up (Median)	Major Findings
STOP-HF [28]	1374 participants with cardiovascular risk factors. Mean age 64.8 years	4.2 years	Lower prevalence of LV dysfunction in the BNP-based screening group (5.3% vs. 8.7%).
PONTIAC [29]	300 patients with T2DM and NT-proBNP > 125 pg/mL but without any history of cardiac disease	2 years	Intensified therapy led to a significant reduction in cardiac hospitalizations/deaths compared to control (hazard ratio: 0.351; *p* = 0.044).
EXAMINE [30]	5380 patients with T2DM and a recent ACS event	597 days	Two NT-proBNP measurements (6 months apart) identified those at the highest risk of developing HF (*p* < 0.001).
CANVAS [31]	4330 patients with T2DM and risk factors for CVD	5.75 years	Higher baseline NT-proBNP levels in those with investigator-reported HF; canagliflozin reduced serial NT-proBNP levels.
Thousand and I [32]	960 patients with T1DM	6.3 years	Increased levels of NT-proBNP associated with worse outcomes (hazard ratio 1.56).

**Table 2 diagnostics-14-02251-t002:** Optimal NP cut-off points according to current guidelines.

Guideline	NP Threshold for Ruling In and Ruling Out HF
AHA [22]	BNP > 35 pg/mL and NT-proBNP > 125 pg/mL to rule in HF in diabetes
APSC [23]	BNP < 35 pg/mL and NT-proBNP < 125 pg/mL to exclude HF
ADA [33]	BNP > 50 pg/mL and NT-proBNP > 125 pg/mL for screening those with diabetes
ESC [34]	BNP > 35 pg/mL and NT-proBNP > 125 pg/mL to identify those with HF

**Table 3 diagnostics-14-02251-t003:** Major HF trials and the role of NP levels in patient selection and outcomes.

Study	Year	Drug Class	NP Level as Selection Criterion	Results
CHARM [54]	2003	ARB	Not a criterion	Non-significant to borderline-significant outcomes
PEP-CHF [55]	2006	ACEI	Not a criterion	Non-significant to borderline-significant outcomes
I-PRESERVE [56]	2008	ARB	Not a criterion	Non-significant to borderline-significant outcomes
PARAGON [57]	2019	ARNI	Optional	Borderline-significant outcomes
EMPEROR-Preserved [58]	2021	SGLT2i	Absolute	Significant outcomes
DELIVER [59]	2022	SGLT2i	Absolute	Significant outcomes

## Data Availability

Not applicable.

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
