# Peer review of "Emerging Role of Natriuretic Peptides in Diabetes Care: A Brief Review of Pertinent Recent Literature"

_diagnostics, 2024, doi:10.3390/diagnostics14192251_

Round 1

Reviewer 1 Report

Comments and Suggestions for Authors

Thank you for inviting me to review a paper by  Dipti Tiwari and Tar Choon Aw.

The paper, interestingly and in a detailed manner, described natriuretic peptides used in the diagnosis of HF in diabetic patients. The authors presented algorithms used both in cardiac and diabetic units. They also presented potential problems with the interpretation of results due to differences in pharmacotherapy. My minor comment concerns ethnicity issues and differences in natriuretic peptides assessment.

Reviewer 2 Report

Comments and Suggestions for Authors

Thanks for the opportunity to review the manuscript entitled ” Emerging Role of Natriuretic Peptides in Diabetes Care: a brief review of pertinent recent Literature”.

I enjoyed reading this manuscript, it is well structured and based on recent research related to interrelation between diabetes mellitus and heart failure. Very interesting was the chapter regarding natriuretic peptides for HF risk stratification and management in diabetes, and the two risk scores developed to predict the risk of HF in type 2 DM individuals.

The authors conclude that NP-guided treatment represent a promising approach for managing heart failure in diabetic patients. In conclusions, I suggest the authors to rephrase and be more specific regarding the expectations for the future of these determinations and the therapy guided by them.

Comments on the Quality of English Language

-

Reviewer 3 Report

Comments and Suggestions for Authors

Dear Editor and Authors,

The title of the review clearly states the role of natriuretic peptides in diabetes care. The abstract section provides an idea of ​​how natriuretic peptides can be used in cardiovascular risk management in diabetic patients. However, it would be useful to expand the abstract to emphasize the main topics in the review, such as the challenges of treatment strategies and future research directions. The power of the data has been increased by using both current and old literature on the subject. However, the inclusion of studies investigating the use and effects of NPs in different populations does not provide the opportunity for comparison in different populations (for example, were there any gender, race, and ethnicity differences in the HF risk assessment of NPs?). This would be useful to know.

The biochemical mechanisms and clinical applications of NPs are discussed in detail. The pathophysiological mechanisms are explained, but clinical studies on the diagnostic and prognostic values ​​of NPs in different patient populations could be added to provide the opportunity for comparison.

The paper uses very few figures and tables. Studies can be summarized in a single table (including their results). The paper can be made easier to understand and read by readers by adding visual elements. In particular, it would be great to have a table summarizing the guidelines' diagnostic algorithms based on monitoring NP levels and NP-based treatment strategies in order to see the differences in the approaches of the guidelines in a single image and to make comparisons.

The paper generally focuses on the role of NPs in diabetes care and touches on an important but relatively less discussed area. It does this by targeting current studies in the literature and summarizing the usage strategies in different studies. However, more clinical comparisons can be made on the subject, and more detailed analysis can be provided by comparing NP-guided treatment strategies with existing clinical studies. In addition, the paper can be strengthened by comparing with other biomarkers other than NPs. At the same time, this helps us to form an idea about which biomarker is more useful in this regard.

Overall, the paper is well-structured and addresses an important topic. However, I believe that the scientific impact of the paper will increase if it is enriched with more visual elements, clinical comparisons, and discussions on future research directions.

Comments on the Quality of English Language

The paper is generally easy to read and fluent, but it is recommended that the English language be reviewed. 
